# A Rare Case of Aortic Dissection 10 Years Post Percutaneous Catheterization Retrieval of an Embolized PDA Device in a Patient with Down Syndrome

**DOI:** 10.3390/pediatric17040084

**Published:** 2025-08-08

**Authors:** Youna Park, Hong Ryang Kil, Sang Yoon Kim, Geena Kim

**Affiliations:** 1Department of Pediatrics, Chungnam National University Sejong Hospital, School of Medicine, Chungnam National University, Daejeon 35105, Republic of Korea; 2Department of Pediatrics, Chungnam National University Hospital, School of Medicine, Chungnam National University, Daejeon 35105, Republic of Korea; 3Department of Thoracic and Cardiovascular Surgery, Seoul National University Bundang Hospital, 172, Dolma-ro, Bundang-gu, Seongnam-si 13620, Gyeonggi-do, Republic of Korea

**Keywords:** aneurysm, aortic dissection, PDA device closure, Down syndrome

## Abstract

There are no recorded cases of catheter-induced aortic dissection in pediatric patients. We report a unique case of a pediatric patient with Down syndrome who developed a long-standing dissecting aortic aneurysm. The patient underwent successful stent insertion 10 years after experiencing difficulty retrieving an embolized patent ductus arteriosus device. The Down syndrome presented a complex clinical scenario, making diagnosis challenging due to a lack of cooperation and uncertainty about when the dissection occurred, as symptoms like pain were not reported. Though rare in children, it is vital to recognize procedures such as percutaneous closure of patent ductus arteriosus followed by device retrieval as potential risk factors for aortic dissections in the pediatric population.

## 1. Introduction

Aortic dissections in children are rare, and their symptoms vary. The cause in children is not well defined; it is most likely congenital and associated with developmental defects such as coarctation of the aorta, bicuspid aortic valve, and Marfan syndrome. Although rare and iatrogenic, such as those encountered during umbilical artery catheterization, the percutaneous closure of various cardiac diseases is also a known cause of aortic dissections [1]. Transcatheter closure of the patent ductus arteriosus (PDA) with an Amplatzer ductal occluder (ADO) (St. Jude Medical Corp., St. Paul, MN, USA) is a well-established procedure with minimal risk; embolization is rare, occurring in approximately 1% of cases. It can affect the pulmonary artery but may also involve the systemic artery, requiring transcatheter or surgical retrieval [2]. We present a rare case of a pediatric patient with Down syndrome (DS) who developed a chronic dissecting aortic aneurysm and successfully underwent stent insertion 10 years after the difficult retrieval of an embolized PDA device.

## 2. Case Description

A two-year-old female patient with DS was first admitted to our hospital for pneumonia in 2011. A concomitant moderately sized PDA with a predominantly left-to-right shunt was noted on admission. Right-heart catheterization revealed a pulmonary blood flow/systemic blood flow of 1.75 and a mean pulmonary artery pressure of 20 mmHg. Therefore, we performed transcatheter PDA closure under general anesthesia. Angiography with a 4 Fr pigtail catheter showed a small PDA with a short conical shape, 0.8 mm in diameter and 0.7 in length, and a 0.1 mm ampulla. A 5/4 mm ADO (St. Jude Medical Corp., St. Paul, MN, USA) was initially selected. The device was successfully implanted and detached using the antegrade approach.

However, after detachment, the catheter accidentally nudged the implanted ADO device, causing embolization of the descending aorta. We initially attempted retrieval through a retrograde approach using a 6 Fr sheath via the left femoral artery with biopsy forceps and snaring; however, this failed because the 6 Fr sheath was too small. The device was eventually retrieved via an antegrade route using an 8 Fr sheath inserted into the left femoral vein. A catheter was then advanced into the descending aorta through the PDA with the aid of forceps. The embolized ADO device was attached to the descending aorta for approximately 2 h. The patient had no hypotension, and oxygen saturation remained static. Subsequently, a 6/4 mm ADO was successfully reimplanted, and the PDA was closed without leakage. The pulses in both lower extremities were weak, and heparin infusion (1 mg/kg/min) was initiated because of a possible thrombus. After the closure of the PDA device, echocardiography revealed no descending aortic obstruction. The patient was successfully discharged and then lost to follow-up.

Ten years after the percutaneous PDA closure, limb blood pressure (BP) discrepancies were noted at the outpatient clinic (upper/extremity [U/E] 162/49 mmHg and lower/extremity [L/E] 131/100 mmHg). The patient was 11 years old, weighed 70 kg, and was 147 cm in height. The echocardiographic findings were as follows: closed PDA with mild aortic regurgitation and obstruction of the abdominal aorta with a Vmax of 2.5 m/s (maximum trans-stenotic pressure gradient of 27 mmHg). Left ventricular function was normal, with an ejection fraction of 64%. Chest radiography was nonspecific, and electrocardiography showed a normal sinus rhythm without specificity. Computed tomography (CT) further confirmed a dissecting descending aorta from T8 to T9 with complete luminal collapse and focal stenosis of the distal descending aorta at the T10 level. The diameter of the descending aorta was 12.9 mm in the upper portion, 5.7 mm in the stenotic portion, and 10.5 mm in the lower portion (Figure 1). Renal function test results remained within the normal ranges.

A metal stenting procedure was performed after a thorough discussion with the cardiothoracic surgical team. Left heart catheterization was performed. Angiography showed stenosis in the descending aorta, with a waist diameter of 8 mm, an upper 20 mm, and a lower 15 mm (Figure 2a). A systolic pressure gradient of 40 mmHg was noted across the stenotic segment (blood pressure above the stenosis, 110/65 mmHg; below the stenosis, 70/60 mmHg). The patient was placed under general anesthesia. First, balloon angioplasty was performed using Powerflex (Powerflex Pro, Cordis, Miami, FL, USA) 6.0 × 80 mm, then 8.0 × 100 mm, and finally Mustang (Boston Scientific, Marlborough, MA, USA) 9.0 × 40 mm. Despite this procedure, residual stenosis with a systolic blood pressure drop greater than 30 mmHg across the stenotic portion remained (blood pressure above the stenosis 110/70 mmHg; below the stenosis). A 12 mm (5.5 cm) Andrastent XXL aortic stent (AndraMed GmbH, Reutlingen, Germany) was deployed (inflated by a balloon at 7 atm pressure) and successfully inserted into the descending aorta (Figure 2b). The trans-stenotic pressure gradient decreased significantly to 5 mmHg. Post-procedure CT showed significant improvement; however, a mild residual waist circumference was present at the T10 level (Figure 3). After stent implantation, the limb BP discrepancies decreased (U/E 120/65 mmHg and L/E 115/56 mmHg). The patient was discharged on postoperative day 3. During the months-long follow-up period in the outpatient setting, the patient had no other complications.

## 3. Discussion

This is the first reported case of a chronic aortic dissection after PDA device retrieval in a pediatric patient. Aortic dissections are rare in pediatric patients, with an incidence of approximately 5–10 cases per million and risk factors including connective tissue disease, congenital cardiac disease, severe trauma, cocaine use, and chronic hypertension. It is associated with a high mortality rate (38%) [1]. Symptoms of aortic dissection include sharp chest pain with back or abdominal pain, pulse deficit, and syncope [3]. Our patient had no symptoms or could not accurately describe them. The onset of aortic dissection was unknown in our case but could have been in a chronic state. Aortic dissection is considered chronic if it persists for >14 days after the initial tear of the aortic artery. Chronic aortic dissection carries the risk of aortic aneurysm and rupture. Early diagnosis is crucial because as blood flows through the false lumen, it aggravates aneurysm dilation and increases the risk of rupture [3].

Down syndrome is not directly associated with aortic dissection, as it does not typically involve connective tissue abnormalities like other genetic disorders such as Marfan syndrome, Loeys–Dietz syndrome, and vascular Ehlers–Danlos syndrome [4,5]. Although, some studies have reported that patients with Down syndrome exhibit a reduced number and impaired function of endothelial progenitor cells, which play a crucial role in vascular repair. This may lead to decreased capacity for endothelial recovery and could be associated with an increased risk of cardiovascular and vascular-related diseases [6]. But, in the present case, the aortic dissection is unlikely to have been caused by the patient’s underlying Down syndrome. Instead, we believe that the dissection may have resulted from mechanical injury associated with the embolization and subsequent retrieval of the PDA device.

While there is a 10-year interval between the device embolization and the diagnosis of aortic dissection, we consider the causal relationship plausible due to the following factors: the dissection was anatomically adjacent to the site of device retrieval, and the patient had no other identifiable risk factors such as hypertension, congenital aortic abnormalities, or connective tissue disorders. Although definitive proof of causality cannot be established in a single case, the anatomical proximity and absence of alternative etiologies support a possible delayed vascular injury related to the prior intervention.

However, the possibility of spontaneous aortic dissection in pediatric patients cannot be completely ruled out. Although extremely rare, cases of spontaneous aortic dissection in children have been reported. In addition, hypertension can be a potential contributing factor to aortic dissection in pediatric patients. In this case, the patient had not been followed up for the past 10 years, making it difficult to determine whether hypertension was present. However, if hypertension had been present, it could have contributed to the progression of intimal injury in an already weakened aortic wall [7].

The embolized PDA device likely damaged the aortic intima during the initial closure procedure. Subsequently, the unfolded device was manipulated with forceps and snared several times when repositioning the device back into the catheter. The device was initially inserted into the catheter through the left femoral artery but could not be removed because of the small sheath size. Therefore, it was placed again in the aorta and reinserted into the catheter via the left femoral vein. This process likely caused repeated damage to the aortic intima. Although aortic dissection was not immediately detected post-procedure, weakened intimal tears formed in the false lumen as the child grew.

In our case, the embolized PDA device was successfully retrieved percutaneously. However, retrieval is sometimes impossible, causing vascular damage or emergency situations; therefore, surgical support is essential. Patients undergoing PDA device closure and retrieval procedures are at risk of developing aortic dissection. Due to their age, children may be unable to articulate their symptoms clearly. Therefore, it is crucial to closely monitor for postprocedural complications, including aortic dissection, during follow-up. Additionally, patients who experience device embolization or other events during the procedure, as in the present case, require careful observation. PDA device closure is generally safe, but complications can include endocarditis, hemolysis due to incomplete closure and embolization, device-induced left pulmonary artery stenosis, and aortic coarctation. Embolization with an Amplatzer device is rare (0–16%) [8]. Device embolization requiring surgery occurs in 1–2% of cases [9]. Aortic dissection following an intervention is extremely rare, especially in pediatric patients. There have been no recorded instances of catheter-induced aortic dissection in pediatric patients. Although not common, late embolization can occur, so continued follow-up is necessary. A case report described a patient who underwent transcatheter PDA closure at the age of two but experienced late device embolization to the main pulmonary artery (MPA) 18 years later, necessitating surgical retrieval. In addition, several similar cases have also been reported [10].

Aortic stent insertion is often performed in children to treat coarctation or postsurgical re-coarctation [11]. In our case, aortic stent implantation successfully treated aortic stenosis. However, this was not the end of treatment. Complications such as growth-induced stent narrowing can arise. Subsequent repositioning may lead to the shortening or fracturing of the stent. Four months after the procedure, the patient is stable and has no complications. However, continuous monitoring for additional complications is essential.

## 4. Conclusions

Although aortic dissection is exceptionally rare in the pediatric population, invasive procedures such as transcatheter closure of a patent ductus arteriosus, particularly when followed by device retrieval, should be regarded as potential iatrogenic contributors and warrant careful consideration in follow-up.

## Figures and Tables

**Figure 1 pediatrrep-17-00084-f001:**
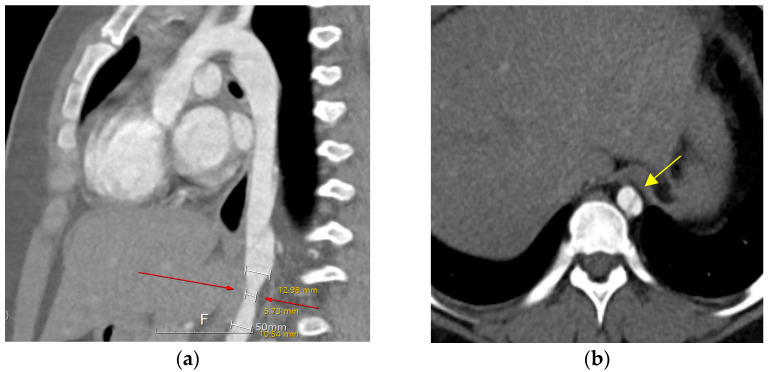
Aorta, abdominal CT. (**a**) Sagittal view, dissection of descending aorta. Focal stenosis at the distal descending aorta at the T10 level (red arrow). (**b**) Axial view, true lumen, and false lumen with intimal flap at T8–9 level (yellow arrow).

**Figure 2 pediatrrep-17-00084-f002:**
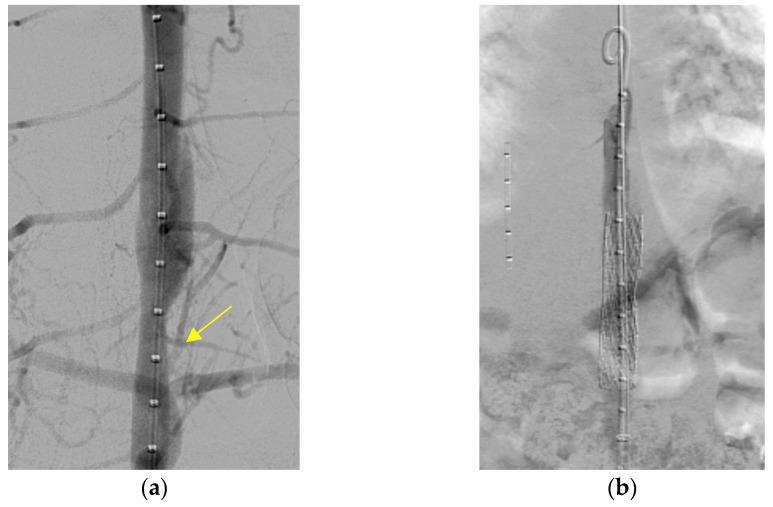
Aorta angiogram and stent insertion. (**a**) An aortic angiogram shows focal stenosis in the descending aorta at the T10 level (yellow arrow). (**b**) A 12/55 mm stent in the aorta was inserted after ballooning.

**Figure 3 pediatrrep-17-00084-f003:**
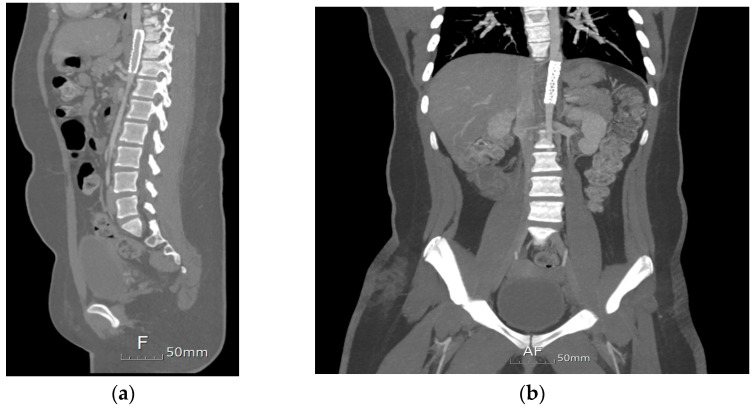
A stent has been placed in the descending aorta at the T9–T11 level. (**a**) Sagittal view (**b**) Coronal view. The focal stenosis of the descending aorta was successfully treated with stent insertion.

## Data Availability

The original contributions presented in this study are included in the article. Further inquiries can be directed at the corresponding author(s).

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
