# Peer review of "A Rare Case of Aortic Dissection 10 Years Post Percutaneous Catheterization Retrieval of an Embolized PDA Device in a Patient with Down Syndrome"

_pediatrrep, 2025, doi:10.3390/pediatric17040084_

Round 1
Reviewer 1 Report
Comments and Suggestions for Authors
I have reviewed the manuscript "A rare case of Aortic dissection, 10 years post percutaneous
catheterization retrieval of an embolized PDA device in a patient with Downs syndrome". It is well-written and easy to follow. My major concern is whether it can actually be confirmed that the 10-year previous retrieval of an embolized patent ductus arteriosus device was the actual cause of the dissecting aortic aneurysm. I am not sure the authors have provided us with sufficient evidence to support this causal relationship.
Author Response
Comment : My major concern is whether it can actually be confirmed that the 10-year previous retrieval of an embolized patent ductus arteriosus device was the actual cause of the dissecting aortic aneurysm. I am not sure the authors have provided us with sufficient evidence to support this causal relationship.
Response : Thank you very much for your thoughtful and critical comment.
We fully acknowledge the reviewer’s concern regarding the causal relationship between the previous device retrieval and the development of the dissecting aortic aneurysm. We agree that, based on a single case report, it is not possible to definitively establish a direct causal link.
However, in our revised manuscript, we have clarified that this relationship remains speculative and is based on the temporal association, anatomical proximity of the lesion to the site of device retrieval, and the absence of other known predisposing conditions for aortic dissection in this patient. We have revised the Discussion section to state this more clearly and emphasized that further studies and case accumulations are required to validate such an association. These considerations have been incorporated into the third paragraph of the Discussion section.
Reviewer 2 Report
Comments and Suggestions for Authors
This is a case report of a patient with Down syndrome who underwent catheter closure of patent ductus arteriosus (PDA) and was diagnosed with aortic dissection 10 years later. This article is valuable, but it needs some modifications.
- The Case Description consists of one paragraph and is difficult to read. I recommend that the description be divided into several paragraphs at appropriate points. For example, “However” in Line 49, “Ten years” in Line 62, and “A metal stenting procedure” in Line 74. Incidentally, it is said that the ideal length of a paragraph is 100 to 200 words and no more than 5 to 6 sentences.
- On the other hand, the Discussion section has seven paragraphs, which is a lot for a case report, and the argument also needs to be organized. The themes of each paragraph are as follows:
#1 Complications of PDA device closure and aortic dissection with catheterization
#2 Intimal damage by PDA devices
#3 This is the first case of chronic aortic dissection after PDA device closure and characteristics
#4 Report of a pediatric case of acute aortic dissection after catheterization
#5 PDA device retrieval with damage
#6 Risk of aortic dissection due to PDA device closure and retrieval technique
#7 Usefulness and caveats of stents for aortic coarctation
The most important paragraph #3 should be the first paragraph of the Discussion. Paragraph #4 could be omitted or mentioned in the first paragraph. Next, paragraphs #2, #5, and #6, which describe the mechanism of aortic dissection in this case, should be combined into the second paragraph. You may omit #1 or leave it as the third paragraph. Since the focus of #7 is on the stent, it should be the third or fourth paragraph without modification.
- Is there any causal relationship between aortic dissection and Down syndrome? Please let us know your opinion.
Author Response
Comment 1: The Case Description consists of one paragraph and is difficult to read. I recommend that the description be divided into several paragraphs at appropriate points. For example, “However” in Line 49, “Ten years” in Line 62, and “A metal stenting procedure” in Line 74. Incidentally, it is said that the ideal length of a paragraph is 100 to 200 words and no more than 5 to 6 sentences.
Response 1: Thank you for your helpful suggestion regarding the structure of the Case Description section. In response, we have revised the Case Description by dividing the content into multiple paragraphs at appropriate transition points. Specifically, we have inserted paragraph breaks around the sentences beginning with “However” (Line 49), “Ten years” (Line 62), and “A metal stenting procedure” (Line 74) as you recommended. We appreciate your guidance in enhancing the readability of our manuscript.
Comment 2: On the other hand, the Discussion section has seven paragraphs, which is a lot for a case report, and the argument also needs to be organized. The themes of each paragraph are as follows:
#1 Complications of PDA device closure and aortic dissection with catheterization
#2 Intimal damage by PDA devices
#3 This is the first case of chronic aortic dissection after PDA device closure and characteristics
#4 Report of a pediatric case of acute aortic dissection after catheterization
#5 PDA device retrieval with damage
#6 Risk of aortic dissection due to PDA device closure and retrieval technique
#7 Usefulness and caveats of stents for aortic coarctation
The most important paragraph #3 should be the first paragraph of the Discussion. Paragraph #4 could be omitted or mentioned in the first paragraph. Next, paragraphs #2, #5, and #6, which describe the mechanism of aortic dissection in this case, should be combined into the second paragraph. You may omit #1 or leave it as the third paragraph. Since the focus of #7 is on the stent, it should be the third or fourth paragraph without modification.
Response 2: Thank you very much for your detailed and constructive feedback regarding the structure of the Discussion section. In response to your suggestion, we have reorganized the Discussion section. We believe that this reorganization has improved the logical flow and clarity of our discussion, in line with your recommendation.
Comment 3: Is there any causal relationship between aortic dissection and Down syndrome? Please let us know your opinion.
Response 3: To our knowledge, Down syndrome is not directly associated with an increased risk of aortic dissection, as the syndrome does not typically involve connective tissue abnormalities that are commonly implicated in aortic wall fragility. In contrast, genetic conditions such as Marfan syndrome, Loeys–Dietz syndrome, vascular Ehlers–Danlos syndrome, and Turner syndrome are well-recognized risk factors for aortic dissection due to their effects on vascular integrity. In this case, we believe that the aortic dissection was not caused by the underlying Down syndrome, but rather may have resulted from mechanical trauma associated with device embolization and retrieval during PDA closure. These considerations have been incorporated into the second paragraph of the Discussion section. Furthermore, the section related to Down syndrome was excluded from the conclusion, as it was deemed not essential to the core discussion of the study
Reviewer 3 Report
Comments and Suggestions for Authors
This single-case study reports on iatrogenic aortic dissection caused by a catheter in a pediatric patient. The case is described as a unique instance of a child with Down syndrome who developed a long-standing aortic dissecting aneurysm, but it is important to note that such complications can occur even in patients without Down syndrome. I have also seen cases of Turner syndrome where patients were transported to the hospital with late-onset aortic dissection.
Although rare in children, procedures such as percutaneous closure of patent ductus arteriosus and subsequent device retrieval may be potential risk factors for aortic dissection in children; however, the data from this single case is insufficient to draw such a conclusion. I request to conduct further research and compile a table of papers related to aortic dissection in pediatric patients.Needs to find more risk factor for aortic dissection.
Author Response
[Reviwer 3]
Comment : This single-case study reports on iatrogenic aortic dissection caused by a catheter in a pediatric patient. The case is described as a unique instance of a child with Down syndrome who developed a long-standing aortic dissecting aneurysm, but it is important to note that such complications can occur even in patients without Down syndrome. I have also seen cases of Turner syndrome where patients were transported to the hospital with late-onset aortic dissection.
Although rare in children, procedures such as percutaneous closure of patent ductus arteriosus and subsequent device retrieval may be potential risk factors for aortic dissection in children; however, the data from this single case is insufficient to draw such a conclusion. I request to conduct further research and compile a table of papers related to aortic dissection in pediatric patients. Needs to find more risk factor for aortic dissection.
Response : Thank you for your valuable comment. In accordance with your suggestion, we acknowledge that the embolization and retrieval process of the PDA device may have played a significant role in the development of the dissecting aortic aneurysm in this case. To clarify this point, we revised the second paragraph of the Discussion section to emphasize that Down syndrome itself is not a direct cause of aortic dissection, and that mechanical injury during the device embolization and retrieval procedure is a more plausible explanation. Despite the 10-year interval between the intervention and the diagnosis, we highlighted the anatomical proximity of the lesion to the site of device retrieval and the absence of other known risk factors in this patient. Additionally, to support this argument with more structured evidence, we included a summary table of previously reported pediatric cases of aortic dissection related to catheter-based or device-related interventions.
Table1 : Pediatric Chronic Aortic Arch Dissection (Catheter Related)
|
Author (year)/Title |
Age/Sex |
Procedure Type/complication |
Key Findings |
|
Yiwei Liu, et al. (2024) Surgical repair of acute aortic dissection concomitant with pulmonary artery dissection due to transcatheter patent ductus arteriosus closure in child |
1year 10 months/Female |
Transcatheter PDA closure/ acute aortic dissection concomitant with pulmonary artery dissection |
While advancing a 6Fr pigtail catheter into the descending aorta via the femoral artery, resistance was encountered. Angiography was performed, which revealed an acute aortic dissection. |
|
Baker & Celermajer (2020) Acute aortic dissection as a late and fatal complication of transcatheter persistent ductus arteriosus occlusion: a case report |
Adult (66/F) |
Transcatheter PDA closure/ Fatal acute aortic dissection after 12 year |
Twelve years after the PDA closure, a large intimal tear was identified adjacent to the PDA occluder. |
|
Wang et al.(2020) Multisite arterial dissection and pseudoaneurysm following umbilical artery catheterization in a neonate |
Neonate/M |
Umbilical arterial catheter placement/ Abdominal aortic dissection, carotid artery dissection
|
An umbilical catheter was inserted on day 1 of life and removed on day 5. Six hours after removal, an abdominal aortic dissection occurred. During subsequent treatment, a carotid artery dissection occurred while attempting a puncture of the left common carotid artery. |
Table2: Pediatric Chronic Aortic Arch Dissection (Non‑Catheter Related)
|
Author (Year) |
Age/Sex/Underlying Condition |
Key Findings |
|
Rathi (2013) Chronic type B aortic dissection in association with Hemolytic uremic syndrome in a child |
9 yr / M / Hemolytic uremic syndrome |
Chronic type B aortic dissection He was started on antihypertensive medication after his first HUS attack, and during a recurrent episode of HUS, he was diagnosed with chronic type B aortic dissection, with poorly controlled blood pressure. |
|
Jo et al. (2011) Aortic dissection and rupture in a child |
11 yr / M / IgA nephropathy three years ago |
Acute type B aortic dissection with rupture The patient was on antihypertensive medication but did not have hypertension. . He presented with sudden chest pain and was diagnosed with aortic dissection. While awaiting surgery, he died due to rupture. |
|
Liang et al. (2022) Management of a 10‑Year‑Old Patient Who Presented With Infective Endocarditis and Stanford Type A Aortic Dissection |
10 yr / F / Infective endocarditis, bicuspid aortic valve
|
Acute Stanford type A aortic dissection A previously healthy 10-year-old female presented to the emergency department with vomiting and headache and was diagnosed with acute Stanford type A aortic dissection. During surgery, infective endocarditis and a bicuspid aortic valve were identified. |
|
Kazimierczak et al. (2018) Endovascular stenting of a complicated type B aortic dissection in an 11‑year‑old patient |
11 yr / F / Previous healthy |
Chronic Stanford Type B aortic dissection with stent repair |
|
Liu et al. (2024) Aortic dissection in children: case report and literature review |
Case1 : 13 yr / M Case2 : 9 yr / M Case3 : 11 yr / M
|
Case1: Stanford Type A Aortic dissection, Polycystic kidney disease type 1 was diagnosed through gene sequencing after surgery for aortic dissection. Case2: Stanford Type A Aortic dissection by accidenct Case3: Stanford Type A Aortic dissection, Loeys-Dietz syndrome was diagnosed through gene sequencing after surgery for aortic dissection.
|
Table: Key Studies on Pediatric Aortic Dissection
|
Authors (Year)/ Title |
Study Type |
Population |
Associated diagnosis |
|
Shamszad et al. (2013) Aortic dissection in hospitalized children and young adults: a multiinstitutional study |
Retrospective administrative cohort |
110 patients <30 yrs with AD (2004–2011) |
Underlying Causes of Aortic Dissection in Children: · Congenital heart disease (CHD): 38% · Trauma: 24% · Connective tissue disease: 16% · Isolated hypertension: 8% Common CHD Diagnoses: · Aortic arch anomalies: 24% · Valve disease: 21% · Hypoplastic left heart syndrome (HLHS): 12% · Transposition of the great arteries (TGA): 7% · Etc : conotruncal canomoalies, PDA Specific Connective Tissue Disorders: · Marfan syndrome: 72% · Ehlers–Danlos syndrome: 11% · Unsepcified CTD : 17% |
|
Hua et al. (2015) Hospital survival of aortic dissection in children |
The Kids' Inpatient Database (1997–2009) |
168 thoracic AD cases <20 yrs |
Hypertension: 18% Marfan syndrome: 14% Aortic valve disorder: 8% Congenital aortic valve insufficiency: 7% Coarctation of the aorta: 5% |
|
Zalzstein et al. (2003) Aortic dissection in children and young adults: diagnosis, patients at risk, and outcomes |
Multi-center chart review (1970–2000) |
13 patients <25 yrs |
Marfan syndrome: 4 cases Coarctation of the aorta: 1 case Tetralogy of Fallot (TOF): 1 case Catheter-related: 2 cases Trauma: 3 cases |

Round 2
Reviewer 3 Report
Comments and Suggestions for Authors
accept in present form
Author Response
We would like to express our sincere gratitude to the reviewers for their thoughtful and constructive comments on our manuscript. We truly appreciate the time and effort you have taken to review our work despite your busy schedules.